# PUTTING MACHINE TRANSLATION IN CONTEXT WITH THE NOISY CHANNEL MODEL

## ABSTRACT

We show that Bayes' rule provides a compelling mechanism for controlling unconditional document language models, using the long-standing challenge of effectively leveraging document context in machine translation. In our formulation, we estimate the probability of a candidate translation as the product of the unconditional probability of the candidate output document and the "reverse translation probability" of translating the candidate output back into the input source language document—the so-called "noisy channel" decomposition. A particular advantage of our model is that it requires only parallel sentences to train, rather than parallel documents, which are not always available. Using a new beam search reranking approximation to solve the decoding problem, we find that document language models outperform language models that assume independence between sentences, and that using either a document or sentence language model outperforms comparable models that directly estimate the translation probability. We obtain the best-published results on the NIST Chinese–English translation task, a standard task for evaluating document translation. Our model also outperforms the benchmark Transformer model by approximately 2.5 BLEU on the WMT19 Chinese–English translation task.

## 1 INTRODUCTION

There have been many recent demonstrations that neural language models based on Transformers (Vaswani et al., 2017; Dai et al., 2019) are capable of learning to generate remarkably coherent documents with few (Zellers et al., 2019) or no (Radford et al., 2019) conditioning variables. Although large unconditional language models are frequently used to provide representations for natural language understanding applications (Devlin et al., 2019; Yang et al., 2019; Peters et al., 2018), how to use them for conditional generation applications remains an open question.

In this paper, we employ Bayes' rule to use powerful unconditional document language models to improve a conditional task: machine translation. The application of Bayes' rule to transform the translation modeling problem $p(\boldsymbol{y} \mid \boldsymbol{x})$, where $\boldsymbol{y}$ is the target language, and $\boldsymbol{x}$ is the source language, has a long tradition and was the dominant paradigm in speech and language processing for many years (Brown et al., 1993). In this context it is known as the noisy channel (Shannon & Weaver, 1949), by analogy to an information theoretic conception of Bayes' rule as a model for reasoning about messages being sent across an unreliable communication medium. In the noisy channel decomposition (§2), rather than directly modeling the conditional distribution, we rewrite it as $p(\boldsymbol{y} \mid \boldsymbol{x}) \propto p(\boldsymbol{y}) \times p(\boldsymbol{x} \mid \boldsymbol{y})$. More importantly, this changes the learning problem from estimating a single complex conditional distribution to learning two different distributions: a language model $p(\boldsymbol{y})$, which provides unconditional estimates of the output (in this paper, documents); and $p(\boldsymbol{x} \mid \boldsymbol{y})$, which provides the probability of translating a candidate output $\boldsymbol{y}$ into the (observed) source $\boldsymbol{x}$.

As we will discuss below, although the problems of estimating $p(\boldsymbol{y} \mid \boldsymbol{x})$ and $p(\boldsymbol{x} \mid \boldsymbol{y})$ are similar, independence assumptions made in $p(\boldsymbol{x} \mid \boldsymbol{y})$ are less statistically costly than they might otherwise be since, at test time, we will be conditioning on $\boldsymbol{x}$ and reasoning about a posterior distribution over $\boldsymbol{y}$, which will be jointly dependent on all (conditionally independent) parts of $\boldsymbol{x}$. This statistical fact—which is the same trick that gives naïve Bayes classifiers their expressiveness and ease of estimation—permits us to assume independence between sentence translations in the reverse translation model, and therefore to use parallel sentences (rather than parallel documents) to train it. In

the posterior, we thus have an implicit estimate of a document-level translation system, even though we made no use of parallel documents. This is particularly useful since parallel sentences are much more readily available than parallel documents. A second benefit of our approach is that the unconditional language model can be estimated from nonparallel data, which exist in vast quantities. The benefits of strong language models estimated from nonparallel data has long been recognized in translation (Brants et al., 2007), and more recently in noisy channel approaches to translation based on neural network component distributions (Yu et al., 2017; Yee et al., 2019; Ng et al., 2019).

Although the noisy channel model is ideal for exploiting the data resources that naturally exist in the world (large corpora of parallel but independent sentences, and large corpora of monolingual documents), we are faced with a much harder decoding problem (§3). To address this problem, we propose a new beam-search algorithm, exploiting the fact that our document language model operates left-to-right, and our reverse translation model treats sentences independently. The search is guided by a proposal distribution that provides candidate continuations of a document prefix, and these are reranked according to the posterior distribution. In particular, we compare two proposal models: one based on estimates of independent sentence translations (Vaswani et al., 2017) and one that conditions on the source document context (Zhang et al., 2018).

To explore the performance of our proposed model, we focus on Chinese–English translation, following a series of papers on document translation (Zhang et al., 2018; Werlen et al., 2018; Tu et al., 2018; Xiong et al., 2019). Although in general it is unreasonable to expect that independent translations of sentences would lead to coherent translations of documents between any pair of languages, the task of translating Chinese into English poses some particularly acute challenges. As Chinese is less morphosyntatically complex than English, and the relevant clues for predicting, e.g., what tense a verb should be in, or whether a noun is singular or plural, may be spread throughout a document, it is crucial that extra sentential context is used.

Our experiments (§4) explore: (1) different approaches to reranking, (2) different independence assumptions when modelling documents (i.e., whether sentences are generated independently or not), (3) different amounts of language modelling data, and (4) different proposal models. Briefly summarized, we find that document-context language models significantly improve the translation quality obtained with our system, both in terms of BLEU scores, and in terms of a human evaluation. Targeted error analysis demonstrates the document prior is capable of enforcing consistency of tense and number and lexical choice across documents.

## 2 MODEL DESCRIPTION

We define $\underline{\boldsymbol{x}} = (\boldsymbol{x}^1, \boldsymbol{x}^2, \ldots, \boldsymbol{x}^I)$ as the source document with $I$ sentences, and similarly, $\underline{\boldsymbol{y}} = (\boldsymbol{y}^1, \boldsymbol{y}^2, \ldots, \boldsymbol{y}^J)$ as the target document with $J$ sentences. During the (human) translation process, translators may split or recombine sentences, but we will assume that $I = J$.[1] Let $\boldsymbol{x}^i = (x_1^i, x_2^i, \ldots, x_M^i)$ represent the $i$th sentence in the document, consisting of $M$ words; likewise $\boldsymbol{y}^i = (y_1^i, y_2^i, \ldots, y_N^i)$ denote the $i$th sentence in the target document, containing $N$ words.

The translation of a document $\underline{\boldsymbol{x}}$ is determined by finding the document $\hat{\underline{\boldsymbol{y}}}$, where $p(\hat{\underline{\boldsymbol{y}}} \mid \underline{\boldsymbol{x}})$ is optimal.

$$\hat{\underline{\boldsymbol{y}}} = \arg\max_{\underline{\boldsymbol{y}}} p(\underline{\boldsymbol{y}} \mid \underline{\boldsymbol{x}}). \tag{1}$$

Instead of modelling the probability $p(\underline{\boldsymbol{y}} \mid \underline{\boldsymbol{x}})$ directly, we factorize it using Bayes' rule:

$$\hat{\underline{\boldsymbol{y}}} = \arg\max_{\underline{\boldsymbol{y}}} \frac{p(\underline{\boldsymbol{x}} \mid \underline{\boldsymbol{y}}) \times p(\underline{\boldsymbol{y}})}{p(\underline{\boldsymbol{x}})} = \arg\max_{\underline{\boldsymbol{y}}} \underbrace{p(\underline{\boldsymbol{x}} \mid \underline{\boldsymbol{y}})}_{\text{channel model}} \times \underbrace{p(\underline{\boldsymbol{y}})}_{\text{language model}}. \tag{2}$$

We further assume that sentences are independently translated, and that the sentences are generated by a left-to-right factorization according to the chain rule. Therefore, we have

$$\hat{\underline{\boldsymbol{y}}} = \arg\max_{\underline{\boldsymbol{y}}} \prod_{i=1}^{|\underline{\boldsymbol{x}}|} p(\boldsymbol{x}^i \mid \boldsymbol{y}^i) \times p(\boldsymbol{y}^i \mid \underline{\boldsymbol{y}}^{<i}), \tag{3}$$

---

[1]Size mismatches are addressed by merging sentences using sentence alignment algorithms.

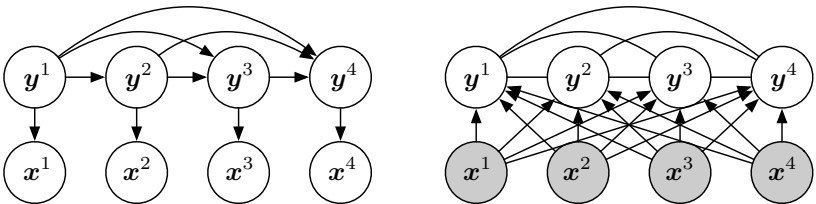

Figure 1: Graphical model showing the factorization of our noisy channel model where $\boldsymbol{y}^i$ indicates the $i$th target language sentence and $\boldsymbol{x}^i$ indicates the $i$th source language sentence. In the prior (left) the target sentences (the $\boldsymbol{y}^i$'s) only influence the corresponding source sentence and therefore can be learned and modeled independently, but at test time (right), when the target is not observed, each $\boldsymbol{y}^i$ is dependent on every $\boldsymbol{x}^i$.

where $\underline{\boldsymbol{y}}^{<i} = (\boldsymbol{y}^1, \dots, \boldsymbol{y}^{i-1})$ denotes a document prefix consisting of the first $i-1$ target sentences. Thus conceived, this is a generative model of parallel documents that makes a particular independence assumption; we illustrate the corresponding graphical model on the left side of Figure 1.

## 2.1 IMPACT OF THE CONDITIONAL INDEPENDENCE ASSUMPTION

At first glance, the conditional independence assumption we have made might seem to be the very independence assumption that bedevils traditional neural document translation systems—translations of sentence $i$ appear to be uninfluenced by the translation of any sentence $j \neq i$. However, while this is the case during training, this is *not* the case at test time. Then, we will be conditioning on the $\boldsymbol{x}_i$'s (the source language sentences), and reasoning about the posterior distribution over the "underlying" $\boldsymbol{y}_i$'s. By conditioning on the child variables, conditional dependencies between all $\boldsymbol{y}_i$'s and between each $\boldsymbol{y}_i$ and all $\boldsymbol{x}_i$'s are created (Shachter, 1998). The (in)dependencies that are present in the posterior distribution are shown in the right part of Figure 1.

Thus, although modelling $p(\boldsymbol{y} \mid \underline{\boldsymbol{x}})$ or $p(\underline{\boldsymbol{x}} \mid \boldsymbol{y})$ would appear to be superficially similar, the statistical impact of making a conditional independence assumption is quite different. This is fortunate, as it makes it straightforward to use parallel sentences, rather than assuming we have parallel documents which are less often available (Voita et al., 2019; Zhang et al., 2018; Maruf et al., 2019, *inter alia*). Finally, since we only need to learn to model the likelihood of sentence translations (rather than document translations), the challenges of guiding the learners to make robust generalizations in direct document translation models (Voita et al., 2019; Zhang et al., 2018; Maruf et al., 2019, *inter alia*) are neatly avoided.

## 2.2 LEARNING

We can parameterize the channel probability $p(\boldsymbol{x}^i \mid \boldsymbol{y}^i)$ using any sequence-to-sequence model and parameterize the language model $p(\boldsymbol{y}^i \mid \underline{\boldsymbol{y}}^{<i})$ using any language model. It is straightforward to learn our model: we simply optimize the channel model and the language model separately on parallel data and monolingual data, respectively. We remark that it is a significant practical advantage of this parameterization that we can retrain the channel and language models independently, for example if we acquire more monolingual data, or use different language models with the same channel model conditioned on the domain of the source text.

## 3 DECODING

Because of the global dependencies in the posterior distribution, decoding in our noisy channel model is computationally complex. On one hand, similarly to the decoding problem faced in standard sequence-to-sequence models, we must search over the space of all possible outputs with a model that makes no Markov assumptions. On the other hand, unlike traditional neural MT models, we have to have a complete guess of $\boldsymbol{y}_i$ before we can reliably compute the score of $p(\boldsymbol{x}_i \mid \boldsymbol{y}_i)$ according to our channel model, making traditional greedy and near-greedy algorithms ineffective. To deal with this issue, we leverage an auxiliary proposal model $q(\underline{\boldsymbol{y}} \mid \underline{\boldsymbol{x}})$ which approximates the posterior distribution using a direct model, and we use this to focus on our search on promising parts of the output space.

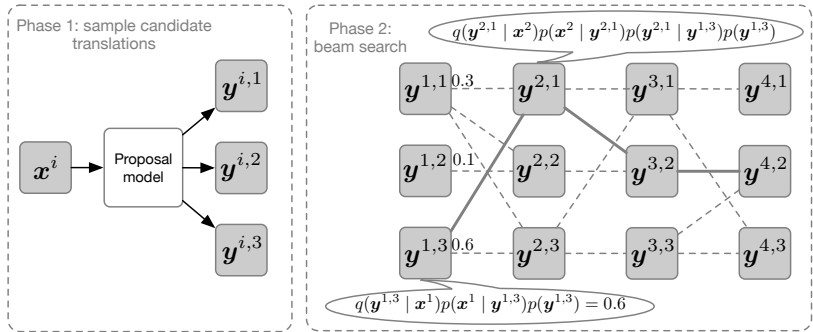

Figure 2: The decoding process. In Phase 1, the auxiliary proposal model generates candidate translations (3 candidates in the diagram) for each sentence in the document (containing 4 sentences). In Phase 2, beam search is employed to search for the best path from the candidate translations.

Because of the autoregressive factorization of the language model, and the independent sentence translation assumption in the channel model, we can carry out the reranking process using a left-to-right beam search strategy with the aid of our proposal model. Figure 2 illustrates the decoding process. For an input document, we let the proposal model propose $K$ candidate translations for each sentence in the document. We then search for the best path from these candidate translations. We use beam search under the objective defined by the following linear model,

$$\mathcal{O}_{\underline{\boldsymbol{x}}^{\leq i}, \underline{\boldsymbol{y}}^{\leq i}} = \lambda_1 \log q(\underline{\boldsymbol{y}}^{\leq i} \mid \underline{\boldsymbol{x}}^{\leq i}) + \lambda_2 \log p(\underline{\boldsymbol{x}}^{\leq i} \mid \underline{\boldsymbol{y}}^{\leq i}) + \log p(\underline{\boldsymbol{y}}^{\leq i}) + \lambda_3 |\underline{\boldsymbol{y}}^{\leq i}|, \qquad (4)$$

where $|\underline{\boldsymbol{y}}^{\leq i}|$ denotes the total number of tokens in the partial document $\underline{\boldsymbol{y}}^{\leq i}$. Note that Equation 4 is a generalization of Equation 2 in $\log$ space—if we set $\lambda_1 = \lambda_3 = 0$ and take the $\log$ of Equation 2 the two objectives are equivalent. The extra factors—the proposal probability and the length of the output—provide improvements (e.g., by calibrating the expected length of the output), and can be incorporated at no cost in the model; they are widely used in prior work (Yu et al., 2017; Yee et al., 2019; Ng et al., 2019).

## 4 EXPERIMENTS

We evaluate our model on two translation tasks, the NIST Chinese–English task and the WMT19 Chinese–English task. On both tasks, we use the standard parallel training data, and compare our model with a strong Transformer baseline, as well as related models from prior work.

### 4.1 DATASET DESCRIPTION

The NIST training data, containing 1.5M sentence pairs, is composed from LDC-distributed news articles and broadcasting scripts. The document-level parallel corpus is a subset of the full training set, including 55K documents with 1.2M sentences. We use the MT06 dataset as validation set and MT03, MT04, MT05, and MT08 as test sets. There are 79 documents and 1649 sentences in the validation set and in total 509 documents and 5146 sentences in the test set. We preprocess the dataset by doing punctuation normalization, tokenization, and lower casing. We use byte pair encoding (Sennrich et al., 2016b) with 32K merges to segment words into sub-word units for both Chinese and English. The evaluation metric is case-insensitive BLEU calculated using `multi-bleu.perl`, which is consistent with prior work on this task.

The training data for the WMT19 Chinese–English task includes the UN corpus, CWMT, and news commentary. The total number of sentence pairs is 18M after filtering the data by removing duplicate sentences and sentences longer than 250 words. The validation sets that we use in the experiment are newstest2017 and newstest2018, which contains 169 documents, 2001 sentences and 275 documents, 3981 sentences, respectively. The test set is newstest2019 containing 163 documents and 2000 sentences. The dataset is preprocessed by segmenting Chinese sentences and normalizing punctuation, tokenizing, and true casing English sentences. As for NIST, we learn a byte pair encoding (Sennrich et al., 2016b) with 32K merges to segment words into sub-word units for both Chinese and English. The evaluation metric is *sacreBLEU* (Post, 2018).

| Method | Model | MT06 | MT03 | MT04 | MT05 | MT08 |
|---|---|---|---|---|---|---|
| (Wang et al., 2017) | RNNsearch | 37.76 | - | - | 36.89 | 27.57 |
| (Kuang et al., 2017) | RNNsearch | - | - | 38.40 | 32.90 | 31.86 |
| (Kuang et al., 2017) | Transformer | 48.14 | 48.05 | 47.91 | 48.53 | 38.38 |
| (Zhang et al., 2018) | Doc-transformer | 49.69 | 50.21 | 49.73 | 49.46 | 39.69 |
| Baseline | Sent-transformer | 47.72 | 47.21 | 49.08 | 46.86 | 40.18 |
|  | Doc-transformer | 49.79 | 49.29 | 50.17 | 48.99 | 41.70 |
|  | Sent-reranker | 51.33 | 52.23 | 52.36 | 51.63 | 43.63 |
| This work | Doc-reranker | **51.99** | **52.77** | **52.84** | **51.84** | **44.17** |

Table 1: Comparison with prior work on NIST Chinese–English translation task. The evaluation metric is tokenized case-insensitive BLEU. The first four rows are numbers reported in the papers of prior work. The first two baselines are the results that we got by running the Transformer (Vaswani et al., 2017) and the Document Transformer (Zhang et al., 2018) on the NIST dataset. The sent-reranker is a variation of our model in which sentences in documents are assumed to be independent.

## 4.2 MODEL CONFIGURATION

For NIST, we use the Transformer (Vaswani et al., 2017) as the channel model and the Document Transformer (Zhang et al., 2018) as the proposal model. The hyperparameters for training the Transformer is the same as *transformer base* (Vaswani et al., 2017), i.e. 512 hidden size, 2048 filter size, 8 attention heads, and 6 layers for both the encoder and decoder. We follow Zhang et al. (2018)'s configuration to train the Document Transformer: context length is set to 2 and all other hyperparameters are the same as *transformer base*. Both models are optimized using Adam (Kingma & Ba, 2015) with approximately 24K BPE tokens per mini-batch. For the language model, we train the Transformer-XL (Dai et al., 2019) on a combination of the English side of NIST training data as well as three sections of Gigaword: XIN, AFP, APW, resulting in a total of 7.3M documents and 115M sentences. We use an architecture with 24 layers, 16 attention heads, and embeddings of dimension 1024. The input sequences to the language model are encoded into bytes using the byte-level encoder provided by GPT2 (Radford et al., 2019).

For WMT19, we use the Transformer (Vaswani et al., 2017) as both the channel and proposal model. The hyperparameters for training the Transformer is the same as *transformer big* (Vaswani et al., 2017), i.e. 1024 hidden size, 4096 filter size, 16 attention heads, and 6 layers. The model is trained on 8 GPUs with batch size of 4096. The setup for the language model is the same as that of NIST except that the training data is the English side of the parallel training data and Gigaword.

For both tasks, the weights $\lambda$ are selected using grid search, from $[0.8, 1., 1.5, 2., 2.2, 2.5, 3.]$ for the weights of channel model $\lambda_2$ and proposal model $\lambda_1$, and from $[0.2, 0.5, 0.8, 1.]$ for the length penalty $\lambda_3$. The size of the $n$-best list used in the reranker is set to 50.

## 4.3 EXPERIMENTAL RESULTS

Table 1 presents the best result from our model (doc-reranker) in comparison with prior work on the NIST Chinese–English translation task. The first four rows are numbers reported in prior work. Wang et al. (2017) incorporate document context by introducing a hierarchical RNN to an LSTM sequence-to-sequence model. Kuang et al. (2017) use a cache to store previously translated words across sentences, which they then use in sequence-to-sequence models. Zhang et al. (2018) extend the Transformer model with an extra context encoder to capture information from previous source sentences. Apart from prior work, we also compare our doc-reranker with three baselines: the Transformer (Vaswani et al., 2017) and Document Transformer (Zhang et al., 2018), and the sentence-level reranker (sent-reranker). In the sent-reranker, we assume sentences in the document are independent (formulation $\hat{\underline{y}} = \arg\max_{\underline{y}} \prod_{i=1}^{|\underline{x}|} p(\underline{x}^i \mid \underline{y}^i) \times p(\underline{y}^i)$), and therefore we train a sentence-level language model and rerank each sentence independently. This sent-reranker setup is close to the work from Yee et al. (2019) and Ng et al. (2019) with the difference that rather than using a language model trained on documents we use a language model trained on sentences, which is more statistically consistent. Table 1 shows that our reranker outperforms previous models as

| Proposal model | Language model | Sent-reranker | Doc-reranker |
|---|---|---|---|
| Sent-transformer | LSTM: NIST | 49.92 | 50.24 |
| | Transformer-XL: NIST | 50.29 | 50.56 |
| | Transformer-XL: NIST + Gigaword | 50.19 | 50.93 |
| Doc-transformer | LSTM: NIST | 50.75 | 51.20 |
| | Transformer-XL: NIST | 51.27 | 51.68 |
| | Transformer-XL: NIST + Gigaword | 51.33 | 51.99 |

Table 2: BLEU scores on NIST dev set MT06 from rerankers which are incorporated with various language models. In the language model column X: Y means the language model X is trained on dataset Y. A bigger language model improves the doc-reranker but does not help the sent-reranker.

| Independence | Architecture | Data | Perplexity |
|---|---|---|---|
| Indep. Sentences | Transformer-XL | NIST | 83.3 |
| | Transformer-XL | NIST + Gigaword | 96.5 |
| Indep. Documents | LSTM | NIST | 71.6 |
| | Transformer-XL | NIST | 43.8 |
| | Transformer-XL | NIST + Gigaword | 43.4 |

Table 3: Perplexity per word (whitespace delimited token) of language models on NIST dev set.

well as strong Transformer baselines by a significant margin—approximately 2.5 BLEU on all test sets—achieving new state of the art. Although the gap between the doc-reranker and sent-reranker is smaller, as we will show in §A.1 and §5 that translations generated by doc-reranker are preferred by humans and are more consistent across documents, in line with concerns about the reliability of using BLEU at assessing cross-sentential consistency (Voita et al., 2019).

To understand the rerankers better, we investigate the effect of different proposal models, different language models, and various numbers of candidates in the $n$-best list. Table 2 and Figure 3 show that better proposal models and bigger $n$-best lists lead to consistently better reranking results. This is an appealing behaviour showing that our reranker is able to pick better translations from higher quality and more diverse candidate pools generated by better proposal models and bigger $n$-best lists. To compare the effect of language models, we train an LSTM language model (Merity et al., 2018a;b) and a Transformer-XL language model on the English side of NIST parallel training data in addition to the Transformer-XL trained on NIST and Gigaword. Table 3 lists the perplexity per word on the NIST validation set for different language models. Given the same training data, the Transformer-XL performs significantly better than the LSTM-based language model, which in turn results in a higher BLEU score from the doc-reranker. By adding more training data, the Transformer-XL language model achieves even lower perplexity and that gives a further boost to the performance of the doc-reranker. Notably, when the strong Transformer-XL language model is incorporated into the doc-reranker, the best weight ratio of the channel and language model is $1 : 1$, indicating that the doc-reranker depends heavily on the language model. By contrast, if a weak language model is incorporated, the best ratio is approximately $2 : 1$. A further observation is that although a larger-scale language model improves the doc-reranker, it does not help the sent-reranker.

We perform an ablation study to explore what each component of the doc-reranker contributes to the overall performance. Table 4 shows BLEU scores on the NIST validation set for the optimal interpolation of various component models. No gains are observed if the language model is combined with the proposal model (a probabilistically unsound combination, although one that often worked in pre-neural approaches to statistical translation). We find that as we increase the weight of the language model, the results become worse. The interpolation of the proposal model and channel model slightly outperforms the proposal model baseline but considerably underperforms the interpolation of the proposal model, channel model, and the language model. This difference indicates the key roles that the language model plays in the doc-reranker. When the channel model is combined with the language model the performance of the doc-reranker is comparable to that with all three compo-

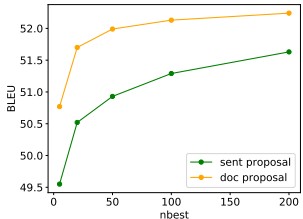

Figure 3: Effect of *n*-best list.

| Reranker | Models | MT06 |
|---|---|---|
| - | Doc-transformer | 49.79 |
| Doc-reranker | Proposal + LM | 49.79 |
| | Channel + LM | 51.93 |
| | Proposal + Channel | 50.40 |
| | Proposal + Channel + LM | **51.99** |

Table 4: Effect of different components in doc-reranker.

| Method | Model | Unpaired Data | LM PPL | Test17 | Test18 | Test19 |
|---|---|---|---|---|---|---|
| Baseline | Transformer big | - | - | 23.9 | 23.9 | 24.5 |
| This work | Doc-reranker | WMT | 106.3 | 24.9 | 26.0 | 27.1 |
| | | Gigaword + WMT | 63.8 | **25.5** | **26.3** | **27.1** |

Table 5: SacreBLEU of different models on WMT19 validation and test sets and perplexity per word of the language models on the English side of WMT19 validation set.

nents included. We conclude from the ablation study that both the channel and language models are indispensable for the doc-reranker, as is consistent with our Bayes' rule account of this model.

Table 5 presents the results of our model together with baselines on the WMT19 Chinese–English translation task. We find that the results follow the same pattern as those on NIST: a better language model leads to better translation results and overall the reranker outperforms the Transformer-big by approximately 2.5 BLEU. The two best systems submitted to the WMT19 Chinese–English translation task are Microsoft Research Asia's system (Xia et al., 2019) and Baidu's system (Sun et al., 2019), both of which employ multiple techniques to improve upon the Transformer big model. Here we mainly compare our results with those from Xia et al. (2019) because we use the same evaluation metric *SacreBLEU* and the same validation and test sets. Using extra parallel training data and the techniques of masked sequence-to-sequence pretraining (Song et al., 2019), sequence-level knowledge distillation (Kim & Rush, 2016), and backtranslation (Edunov et al., 2018), the best model from (Xia et al., 2019) achieves 30.8, 30.9, and 39.3 on newstest2017, newstest2018, and newstest2019, respectively. Although our best results are lower than this, it is notable that our reranker achieves comparable results to their model trained on 56M parallel data, more than two times more training data than we use. However, our method is orthogonal to these works and can be combined with other techniques to make further improvement.

## 5 ANALYSIS

To investigate how the rerankers improve translation quality, we analyze the output from different models: the Document Transformer (Zhang et al., 2018) (our proposal model), the sent-reranker, and the doc-reranker. We observe that in general the doc-reranker improves adequacy of translations and can generate more fluent and natural sentences compared to the Document Transformer. More importantly, our doc-reranker shows its superiority over the others in terms of exploiting context, improving consistency of tense, number, and lexical choice across entire articles. Tables 6 and 7 in Appendix A present example output from the aforementioned systems. In Example 1, the pronoun *he* is omitted in the Chinese sentence. While the Document Transformer misses this pronoun resulting in a translation of completely different meaning, the doc-reranker is able to recover it. Likewise, in Example 7 *them* is dropped in the source sentence and this pronoun can only be inferred from previous context. Although both rerankers recover some pronoun, only the doc-reranker gets it right, by relying on cross-sentential context. Example 2 is a good example showing that the doc-reranker is better at generating adequate translations than the proposal model: the Document Transformer ignores the phrases *with these people* and *at present*, but the doc-reranker covers them.

Chinese does not mark grammatical number, it therefore has to be inferred for the English translations. It is not possible for a sentence-level MT system to capture this information if the relevant context is not from the current sentence. In Example 4 and 6 the plural *problems* and *identities* can only be inferred from previous sentences (the immediate previous sentence in Example 4 and the sentence 4-5 sentences away from the current one in Example 6). While neither the Document Transformer nor the sent-reranker makes the right predictions in both examples, the doc-reranker translates correctly, indicating its strength in capturing extra-sentential information. In addition to making inference across sentences, the doc-reranker is also capable of maintaining consistency of tense and lexical choice, as demonstrated in Examples 5, 8, and 10. Furthermore, impressively, it improves the consistency of writing style. To illustrate, in Example 9, the context is that of a list of bullet points starting with *continue*. The doc-reranker follows in this style by starting the translation with the verb *continue*. However, the sent-reranker starts the sentence with *we should continue*. Although both translations are reasonable, the former one is more natural within the document.

## 6    RELATED WORK

Our work is closely related to three lines of research: context-aware neural machine translation, large-scale language models for language understanding, and semi-supervised machine translation. Recent studies (Tiedemann & Scherrer, 2017; Bawden et al., 2018, *inter alia*) have shown that exploiting document-level context improves translation performance, and in particular improves lexical consistency and coherence of the translated text. Existing work in the area of context-aware NMT typically adapts the MT system to take additional context as input, either a few previous sentences (Jean et al., 2017; Wang et al., 2017; Tu et al., 2018; Titov et al., 2018; Zhang et al., 2018; Werlen et al., 2018) or the full document (Haffari & Maruf, 2018; Maruf et al., 2019). The idea of adding monolingual document-level data has been explored in two most recent works (Voita et al., 2019; Junczys-Dowmunt, 2019). Both use backtranslation (Edunov et al., 2018; Sennrich et al., 2016a) to create synthetic parallel documents as additional training data. We train a large-scale language model and use it to refine the consistency between sentences under a noisy channel framework. An advantage of our model over backtranslation is that the language model is portable across domain and language pairs.

Investigating ways of adding monolingual data to MT systems is an active research area (Gülçehre et al., 2015; Cheng et al., 2016, *inter alia*). Backtranslation (Edunov et al., 2018; Sennrich et al., 2016a), originally invented for semi-supervised MT, has been employed as a standard approach for unsupervised MT (Lample et al., 2018b;a; Artetxe et al., 2019; 2018). Noisy channel decompositions have been a standard approach in statistical machine translation (Brown et al., 1993; Koehn et al., 2007) and recently have been applied to neural models (Yu et al., 2017; Yee et al., 2019; Ng et al., 2019). Unlike prior work, we adopt noisy channel models for document-level MT where language models are responsible for capturing cross-sentence context.

Large-scale pretrained language models have achieved great success in improving systems in language understanding, leading to state-of-the-art results on a wide range of tasks (Peters et al., 2018; Devlin et al., 2019; Radford et al., 2018; McCann et al., 2017; Yang et al., 2019; Chronopoulou et al., 2019; Lample & Conneau, 2019). Language generation is another area where pretrained language models have been applied to, with existing work focusing on fine-tuning approaches for repurposing an unconditional language model (Zhang et al., 2019; Edunov et al., 2019; Song et al., 2019; Dong et al., 2019; Ziegler et al., 2019; de Oliveira & Rodrigo, 2019). In contrast to our work which uses probabilities from language models, that work uses their internal representations.

## 7    CONCLUSION

We have presented a noisy channel reranker and empirically validated it on Chinese–English document-level translation tasks. The noisy channel formulation requires only parallel sentences (rather than documents) but we can use abundant monolingual documents to train the language model component. Experiments show that our proposed model considerably improves translation quality—it achieves approximately 2.5 BLEU higher than transformer baselines. Subjective evaluation further confirms that the language model helps enforce consistency of tense, number, and lexical choice across documents.

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

# A APPENDIX

## A.1 HUMAN EVALUATION

We selected 50 translation triplets (reference translation, translation from the doc-reranker, translation from the sent-reranker) sampled from the validation and test sets of NIST for evaluation by 4 native English speakers. The samples are selected by taking the triplets where the output from the sent-reranker and the doc-reranker have a translation edit rate (Snover et al., 2006) above 17.5%.

For each of these documents, they were presented with a reference translation and with two translations labelled A and B, one generated by the doc-reranker and one generated by the sent-reranker. They were tasked with indicating which of these two they found better overall, considering fluency, idiomaticness and correctness (relatively to the reference).

Each of the human evaluators preferred a majority of doc-reranker translations. When aggregated for each document by majority vote, the doc-reranker translations were considered better in 25 documents, worse for 13, and tied for 12. This is a statistically significant preference at $p < 0.05$ according to an exact one-tailed Binomial test ($n = 38$).

## A.2 COMPARISON OF OUTPUT FROM DIFFERENT SYSTEMS

To investigate how the rerankers improve translation quality, we manually inspect the output from three different systems: the Document Transformer (Zhang et al., 2018), the sent-reranker, and the doc-reranker. Tables 6 and 7 present the comparison between the output from the Document Transformer (Zhang et al., 2018) and sent-reranker and between the output from sent-reranker and doc-reranker, respectively. In general, we find that the doc-reranker outperforms other systems in terms of maintaining consistency of tense, number, and lexical choices across documents. For detailed analysis, we refer readers to §5.

| | |
|---|---|
| **Example 1:** | |
| **source:** | 霍夫曼在接受美国哥伦比亚广播公司新闻杂志「六十分钟」访问时轻叹,那段时期为了得到毒品和酒,真是不择手段。 |
| **reference:** | in an interview on us cbs news magazine 60 minutes, hoffman softly sighed that in such period he would truly do anything to get drugs and alcohol. |
| **transformer:** | in an interview with the cbs news magazine "60 minutes", hoffmann sighed that those days were really unscrupulous in getting drugs and alcohol. |
| **reranker:** | in an interview with the cbs news magazine "60 minutes", hoffmann sighed that at that time in order to obtain drugs and alcohol, he was really unscrupulous. |
| **Example 2:** | |
| **source:** | 与此同时，与这几个人在同一工地工作的十多名中方人员也已经撤回到卡拉奇，目前他们的情绪稳定。 |
| **reference:** | in the meantime, more than 10 chinese personnel working in the same place with these people have been called back to karachi. at present they are emotionally stabilized. |
| **transformer:** | at the same time, more than ten chinese personnel working at the same site have also withdrawn to karachi. their sentiments are now stable. |
| **reranker:** | at the same time, more than ten chinese personnel working with these people on the same site have also withdrawn to karachi. at present, their sentiments are stable. |
| **Example 3:** | |
| **source:** | 中国是世界航海大国,在众多的节日中没有航海节是不应该的。 |
| **reference:** | china is a world navigation power and it would not be proper to lack a navigation festival when the country has so many other celebrations. |
| **transformer:** | china is a major navigation country in the world. it is not appropriate to have no maritime festival during the many holidays. |
| **reranker:** | china is a major maritime country in the world, and it is not appropriate to have no maritime festival during the many festivals. |
| **Example 4:** | |
| **source:** | 基本的问题是什么呢? |
| **context:** | . . . however, legislator yeung, i wish to tell you what i am doing today is to ensure every matter can proceed smoothly after the political review starts. therefore, we have to solve some basic problems first and this is a different thing all together. |
| **reference:** | what are the basic problems? |
| **transformer:** | what is the basic problem? |
| **reranker:** | what are the basic questions? |
| **Example 5:** | |
| **source:** | 作者：正义之剑 |
| **context:** | sword of justice: prospects for 2006 |
| **reference:** | author: sword of justice |
| **transformer:** | author: the sword of righteousness |
| **reranker:** | author: the sword of justice |

Table 6: Example outputs from the Document Transformer (proposal model) and our doc-reranker.

| | |
|---|---|
| **Example 6:** | |
| **source:** | 同时我们在国内用最短的时间，核实清楚了死亡人员的身份。 |
| **context:** | . . . the criminal used a submachine gun to fire a barrage of shots, and three engineers died unfortunately. . . . |
| **reference:** | at the same time, we in china verified the identities of the dead within the shortest possible time. |
| **sent reranker:** | at the same time, we spent the shortest time in china to verify the identity of the deceased. |
| **doc reranker:** | at the same time, we spent the shortest time in china to verify the identities of the deceased. |
| **Example 7:** | |
| **source:** | 现在又要平安的送到家里。 |
| **context:** | . . . when the plane carrying the three survivors and 11 other personnel arrived in Hefei, people waiting at the airport heaved a long sigh of relief. . . . after the incident occurred, it made proper arrangements for them. |
| **reference:** | now they will also be escorted home safely. |
| **sent reranker:** | now they have to send it home safely. |
| **doc reranker:** | now they want to send them safely to their homes. |
| **Example 8:** | |
| **source:** | 哈尔滨往双城方向部分公路封闭, 很多车辆绕行。 |
| **context:** | . . . a traffic accident occurred at the 58 kilometer point of the beijing-harbin highway, with a spill from an oil tanker leading to the closure of a section of the highway. . . . it was learned that the oil tanker contained waste oil from charcoal production. . . . |
| **reference:** | the section of the highway from harbin to shuangcheng was closed, with many vehicles detoured. |
| **sent reranker:** | part of the roads heading towards shuangcheng in harbin are closed, and many vehicles are bypassing. |
| **doc reranker:** | part of the road from harbin to shuangcheng was closed , and many vehicles were bypassing. |
| **Example 9:** | |
| **source:** | —继续打好整顿关闭攻坚战。 |
| **context:** | . . . with regard to coalmine safety this year, saws will effectively carry out the following three tasks: –continue to effectively tackle the tough issue of controlling methane. . . . |
| **reference:** | – continue to effectively tackle the tough issue of restructuring and shutting down. |
| **sent reranker:** | – we should continue to make a success of the rectification and closure battle. |
| **doc reranker:** | – continue to fight the battle of rectification and closure. |
| **Example 10:** | |
| **source:** | 其次,"限额"限制了中美经贸关系的良好发展势头。 |
| **context:** | . . . first, such abuse of "quota" restricts the thorough implementation of world trade organization's free trade principle. on one hand, u.s. is talking in high-sounding tone about "free trade". on the other hand, it re-establishes trade barriers and stabs your back at will with "quotas". does it appear too arbitrary and unfair? |
| **reference:** | second, "quota" limits the nice growth trend in sino-america trade relation. |
| **sent reranker:** | second, the "restriction" restricts the good development momentum of sino-us economic and trade relations. |
| **doc reranker:** | second, the "quota" restricts the good development momentum of sino-us economic and trade relations. |

Table 7: Example outputs from the sent-reranker and the doc-reranker.