# OpenReview forum: "Putting Machine Translation in Context with the Noisy Channel Model"
_ICLR.cc/2020/Conference — Reject_

### Official Review · AnonReviewer1 · 2019-10-19
**Official Blind Review #1**

**Rating:** 3

**Review:**

** Paper summary **
In this paper, the authors propose a new re-ranking mechanism leveraging document-level information. Let X and Y denote two languages for ease of reference. The authors focus on X->Y translation and Y->X is a model used for re-ranking. Specifically,
(1)	Two translation models X->Y and Y->X are trained, where X->Y is a document Transformer and Y->X is a sentence transformer.
(2)	Train a language model P(Y) on document-level corpus (rather than sentence-level LM).
(3)	Given a document with $I$ sentences (x^1, …, x^I), translate each source sentence $x^i$ to K candidates.
(4)	Using beam search guided by Eqn.(4) to find optimal translation paths, which is a combination of X->Y translation, Y->X translation, document-level language model and the number of words.
The authors work on NIST Chinese-to-English translation and WMT’19 Zh->En translation to verify the proposed algorithm.

** Novelty **
The novelty is limited. Compared to the paper “the Neural Noisy Channel” (Yu et. al, 2017), the authors use document Transformer and document-level LM for re-ranking, which is of limited novelty.

** Details **
1.	Some baselines are missing from this paper: (A) dual inference baseline [ref1]; (B) X->Y is sentence-level transformer and LM is sentence-level LM, i.e., (Yu et. al, 2017), where P(Y|X) and P(X|Y) are sentence-level translation models.
2.	In Table 1, the improvement of doc-reranker is not very significant compared to sent-reranker, ranging from 0.21 to 0.66.
3.  In Table 4, “Channel + LM” and "Proposal + Channel + LM" achieved almost the same results. Does it mean that the "proposal" component do not work?
4.	Many models are used in this framework. I am not sure whether simple re-ranking or ensemble can outperform this baseline, e.g., 2 Zh->En + 1 En->Zh

[ref1] Dual Inference for Machine Learning, Yingce Xia, Jiang Bian, Tao Qin, Nenghai Yu, Tie-Yan Liu, IJCAI’17

**Experience Assessment:**

I have published in this field for several years.

**Review Assessment: Checking Correctness Of Derivations And Theory:**

I assessed the sensibility of the derivations and theory.

**Review Assessment: Checking Correctness Of Experiments:**

I carefully checked the experiments.

**Review Assessment: Thoroughness In Paper Reading:**

I read the paper thoroughly.

---

> ### Author Response · Authors · 2019-11-08
> **Response**
>
> Thank you for your review.
>
> Regarding the perceived lack of novelty to Yu et al., 2017, please see the response to Reviewer 2.
>
> Regarding baselines. We do have sentence-level LM results + sentence level proposal results (see Table 2). Regarding the Xia et al. (2017) method — which we argue would be a benchmark, not a baseline — we have provided some back translation results (see response to Reviewer 3), and we can add to the paper. Since back translation techniques are closely related, it does not seem important to add these techniques, especially since the Xia et al. method has not yet been established for document level translation (although it no doubt could be used for this), further it is not obvious that it would provide a convenient way to exploit the kind of data that we wish to exploit (monolingual documents, parallel sentences). The goal of this paper, and why readers should read it, is using a theoretically motivated approach to the document translation problem that solves a central data problem in MT, and characterizing its performance relative to some representative baselines.
>
> Regarding the difference in performance between document and sentence LMs. As we discuss in the paper, with citations to much prior work, the impact of fixing problems related to cross-sentence consistency has a minimal impact on BLEU, but the impact on human judgments can be much more significant. For this reason we also carried out a human evaluation, where the document reranker was favoured two-to-one by our evaluators.
>
> Regarding why the proposal model adds no value to the objective. The fact that the proposal model does not add new information to the objective is expected if Bayes rule yields a better estimate of the translation probability than its direct estimation (i.e., the proposal model). Thus, since we believe our component models (channel and language model) to be well estimated, we expected this redundant component to add no value, and we see this result as a confirmation that Bayesian arguments are trustworthy in this domain (deviations could be expected for a variety of reasons: e.g., poorly calibrated probability distributions, that parameters are chosen to maximize BLEU not to minimize the cross entropy under the posterior distribution). We will clarify this point in the paper.
>
> Regarding the effects of ensembling. We have compare ensembling 2 zh->en + 1 en->zh models:
>
> +------------------------------------------------------------------------------------------+------------+
> |                                     Model                                                                      | MT06     |
> +------------------------------------------------------------------------------------------+------------+
> | ensemble                                                                                                   | 50.04      |
> +------------------------------------------------------------------------------------------+------------+
> | Sent-reranker (sent-level transformer as the proposal model)       | 50.29     |
> +------------------------------------------------------------------------------------------+------------+
> | Doc-reranker (sent-level transformer as the proposal model)        | 50.93     |
> +------------------------------------------------------------------------------------------+------------+
>
> The noisy channel approach outperforms ensembling in BLEU, as has been discussed in previous work on noisy channel approaches that show this isn’t just an effect of ensembling. Also, notably, ensembling sentence level models will not address the document translation problem, nor will it enable us to use monolingual text data, which are two benefits that our technique has.

---

### Official Review · AnonReviewer3 · 2019-10-23
**Official Blind Review #3**

**Rating:** 3

**Review:**

This paper presents a simple approach for document-level machine translation. The idea is to use a language model on the target side and a reverse translation model to choose the best document-level translation. This is theoretically justified by Bayes’ rule and the assumption that the sentences are conditionally independent. The authors implement this idea using a reranking model that rescores 50 candidate translations generated by a standard Transformer model for forward translation.

This is interesting work and the experimental results demonstrate the effectiveness of the approach. However, I am concerned about the (missing) comparison between the proposed approach and the approach that combines backtranslation and a document-level translator (e.g.  Doc-transformer). It seems to me that one could backtranslate a large monolingual corpus and use the resulting parallel documents as additional training data for a document-level translation model. How does the proposed approach compare to such a backtranslation approach?

Another concern is the speed of translation. It seems to me that the computational cost required for generating 50 candidates and reranking them is quite high. I would like to see some experimental results on the actual speed of translation. The aforementioned backtranslation approach should not have this problem, which also makes me unsure about the usefulness of the proposed approach in practice.

**Experience Assessment:**

I have published in this field for several years.

**Review Assessment: Checking Correctness Of Derivations And Theory:**

I assessed the sensibility of the derivations and theory.

**Review Assessment: Checking Correctness Of Experiments:**

I assessed the sensibility of the experiments.

**Review Assessment: Thoroughness In Paper Reading:**

I read the paper at least twice and used my best judgement in assessing the paper.

---

> ### Author Response · Authors · 2019-11-08
> **Response-part 1**
>
> Thank you for your review.
>
> Regarding the question about circumventing the data problem with back-translated documents. While this is a good idea, and there is evidence that it can work well (Junczys-Dowmunt, 2019), it is challenging to train such models well, whereas our model involves straightforward training procedures. Specifically, for back translation to succeed, monolingual data that will be back translated must be carefully selected, and, for good performance, you should filter likely bad translations; the ratio of back translated data and “real” data must be balanced, etc. While techniques for doing this are fairly well established for single sentence models, no such established techniques exist for documents. We do have several results that we can add to the paper which we discuss here to convince you that our results are both interesting and “good”.
>
> First, we did attempt to replicate the technique of Junczys-Dowmunt (2019), but found that in Chinese-English, it was difficult to learn a model that reliably generates the correct number of sentences (contra his findings), which makes a fair comparison challenging. But, to give some calibration for the relative power of back translation vs noisy channel modeling, we did generate a sentence-level proposal model on back translation and compare it to the performance of a sentence-level proposal model trained only on “real” parallel data:
>
> +------+-------------------------------------------------------+---------+---------+---------+---------+---------+
> |        |                        Model                                     | MT06 | MT03 | MT04 | MT05 | MT08 |
> +------+-------------------------------------------------------+---------+---------+---------+---------+---------+
> | 1     | Transformer baseline (q)                          | 49.40  | 49.42 | 50.11 | 48.76  | 41.58 |
> +------+-------------------------------------------------------+---------+---------+---------+---------+---------+
> | 2     | Backtranslation (q')                                    | 51.11 | 52.12 | 51.82 | 51.10  | 43.15 |
> +------+-------------------------------------------------------+---------+---------+---------+---------+---------+
> | 3     | Sent-reranker (using q as proposal)       | 52.25  | 52.21 | 52.35 | 51.28 | 44.27 |
> +------+-------------------------------------------------------+---------+---------+---------+---------+---------+
> | 4     | Doc-reranker (using q as proposal)        | 52.70  | 52.47 | 52.52 | 51.49 | 44.43 |
> +------+-------------------------------------------------------+---------+---------+---------+---------+---------+
> |        | Sent-reranker + back translated              |            |            |            |            |            |
> | 5     | proposal (using q' as proposal)               |  52.95|  53.93 | 53.69 | 53.61  | 45.18 |
> +------+-------------------------------------------------------+---------+---------+---------+---------+---------+
> |        | Doc-reranker + back translated               |            |            |            |            |            |
> | 6     | proposal (using q' as proposal)               |  53.56|  54.80 | 53.94 | 53.86  | 45.85 |
> +------+-------------------------------------------------------+---------+---------+---------+---------+---------+
>
> From these results, we see that while both techniques improve translation, i.e., both (2) and (3) are better than (1), sentence level back translation (2) is less effective than a noisy channel model reranker is (row 3), and, as we showed in the reviewed draft the doc-reranker is better again (row 4). Since we have a new model q’, we can use it as a proposal model for our noisy channel reranker — effectively using the monolingual data twice. Happily, this improves results even further (rows 5-6). Thus, in addition to the challenges of making back translation work at all which we believe argues for the value of our model, we have evidence that (a) the noisy channel approach makes better use of monolingual data than back translation does; (b) using our inference strategy based on reranking samples from a proposal model, samples from a backtranslation-trained proposal model (q’) can be improved further still, providing further evidence that the noisy channel model is well calibrated across a variety of qualities and that it picks up different things than backtranslation does. These results will be broadly of interest to the community, even if we haven’t explored all imaginable back translation configurations. If you have a specific result that you think is particularly important to make this paper acceptable, please identify it, and we will run the comparison.

---

> ### Author Response · Authors · 2019-11-08
> **Response-part 2**
>
> Regarding speed. Search is indeed a hard problem in our model. We intend this paper to ask whether a well-motivated model performs well, and provide a reasonable (if imperfect) inference method. We show that it does work well. Now subsequent work can answer the question of how to make decoding fast. But search is a hard problem that has applications in many areas beyond translation, so this paper adds value to those who would work on this problem. We ourselves intend to work on this now that we know this model is effective, but we also argue that this is a good time to publish these results: others may be interested in knowing about yet another interesting search problem. We will clarify this.

---

### Official Review · AnonReviewer2 · 2019-10-24
**Official Blind Review #2**

**Rating:** 6

**Review:**

Summary:
The paper describes a noisy channel approach for document-level translation, which does not rely on parallel documents to train. The approach relies on a sentence-level translation model (from target-to-source languages) and a document level language model (on target language), each is trained separately. For decoding, the paper relies on another proposal model (i.e., a sentence level translation model from source to target) and performs beam-search weighted by a linear combination of the scores of all three models. Experiments show strong results on two standard translation benchmarks.

Comments:
-  The proposed approach is strongly based on the neural noisy channel model of Yu et al. 2017 but mainly extends it to context aware translation. While the paper is referenced, I believe more emphasis should be put on the differences of the proposed approach
-  It seems that the Document Transformer uses parallel-documents to train, so I am wondering if you can still claim that your approach does not require parallel documents.
-  In general, I think the paper is well written and results are compelling.


**Experience Assessment:**

I have published one or two papers in this area.

**Review Assessment: Checking Correctness Of Derivations And Theory:**

I assessed the sensibility of the derivations and theory.

**Review Assessment: Checking Correctness Of Experiments:**

I assessed the sensibility of the experiments.

**Review Assessment: Thoroughness In Paper Reading:**

I read the paper at least twice and used my best judgement in assessing the paper.

---

> ### Author Response · Authors · 2019-11-08
> **Response**
>
> Thank you for your review.
>
> Regarding the differences to Yu et al., 2017. While both papers indeed use a noisy channel decomposition, the novelty in this paper is the theoretical justification for training a model using only parallel sentences and monolingual documents, and then using it to infer document translations (an important task!). This asymmetry in available data is exactly the situation that exists in the world today, and our model, which addresses it directly and elegantly, will undoubtedly be of general interest. Moreover, while the Yu et al., 2017 model could be used on documents by concatenating their sentences to form a single long sequence, this would not let us use the conditional sentence independence assumptions that gives our model the flexibility to use just parallel sentences. Secondarily, the Yu et al. inference algorithm is specialized to their channel model, and it has a quadratic (in the length of the sentence) complexity, which would be prohibitive for sequences longer than a single sentence; in practice our inference technique is much faster. We will clarify these differences in the paper.
>
> Regarding whether our approach really needs parallel documents. First, there are two models in this paper- the joint translation model and proposal model we use to do inference. The joint translation model is only ever trained using parallel sentences. For inference, we use a proposal model that approximates the posterior, and we compare two variants: one that is trained using just parallel sentences (effectively, we assume independence between translations given the source document) and one that is trained with document context (see Table 2). As predicted, a proposal model that more closely matches the true posterior (i.e., the one with document context) is more effective than one that is less accurate (no document context), but the crucial result is that in both cases, document information has a positive impact on the performance of the system. The secondary result is that search is a hard problem, and while usable approximations exist, this is an important open question. We will clarify this.

---

> > ### Comment · AnonReviewer2 · 2019-11-14
> > **Thank you**
> >
> > Thanks for your clarifications.

---

### Decision · Program_Chairs · 2019-12-19

**Decision:**

Reject

**Comment:**

The authors propose using a noisy channel formulation which allows them to combine a sentence level target-source translation model with a language model trained over target side document-level information. They use reranking of a 50-best list generated by a standard Transformer model for forward translation and show reasonably strong results.  The reviewers were concerned about the efficiency of this approach and the limited novelty as compared to the sentence-level noisy channel research Yu et al. 2017. The authors responded in depth, adding results with another baseline which includes backtranslated data. I feel that although this paper is interesting, it is not compelling enough for inclusion in ICLR.